# Exploring the Health Benefits of Yeast Isolated from Traditional Fermented Foods in Korea: Anti-Inflammatory and Functional Properties of *Saccharomyces* and Non-*Saccharomyces* Strains

**DOI:** 10.3390/microorganisms11061503

**Published:** 2023-06-05

**Authors:** Woo-Soo Jeong, Ha-Ram Kong, So-Young Kim, Soo-Hwan Yeo

**Affiliations:** Fermented and Processed Food Science Division, Department of Agrofood Resources, Rural Development Administration, National Institute of Agricultural Sciences, Wanju 55365, Republic of Korea; wjddntnek@korea.kr (W.-S.J.); ohr337@korea.kr (H.-R.K.); foodksy@korea.kr (S.-Y.K.)

**Keywords:** *Saccharomyces*, non-*Saccharomyces*, anti-inflammatory activity, functional property

## Abstract

Traditional yeast (*Saccharomyces cerevisiae*) has been used for its benefits in various fermentation processes; the benefits of non-*Saccharomyces* yeast as a material for food, feed, and pharmaceuticals have been studied recently. This study evaluated the anti-inflammatory activity and extracellular functional characteristics of wild-type yeasts isolated from traditional fermented foods (*doenjang* (common name: soybean paste) and *nuruk*) in Korea. The viability of the yeast and lipopolysaccharide (LPS)-stimulated RAWBlue™ cells was improved, similar to unstimulated RAWBlue™ cells, and the isolates demonstrated NF-κB inhibitory activity. Yeast suppressed the nitric oxide production in LPS-stimulated RAWBlue™ cells, which was attributed to the inhibition of iNOS or COX-2 mRNA expression depending on the strain. Although there were differences depending on the strain, the production of anti-inflammatory cytokines was reduced in the yeast and LPS-stimulated RAWBlue™ cells, some of which were demonstrated at the mRNA level. In addition, the isolates exhibited high antioxidant and antihypertensive activities (similar to the positive control), which varied depending on the strain. This suggests that yeast can be used for fermentation with enhanced antioxidant and antihypertensive activities. Furthermore, the isolates inhibited the growth of pathogenic Gram-negative bacteria, indicating that yeast can inhibit food spoilage and the growth of pathogenic bacteria during fermentation. Consequently, utilizing raw materials to cultivate yeast strains could be a promising avenue for developing functional foods to prevent and treat inflammatory reactions; such foods may exhibit antioxidant, antihypertensive, and antibacterial properties.

## 1. Introduction

Yeast, a microorganism generally distributed in food and soil, possesses several beneficial physiological functions [1]. During food fermentation, yeast plays crucial roles in alcohol and organic acid production, improving flavor and aroma, enhancing nutritional characteristics, and reducing anti-nutritive components and toxins [2,3]. In addition, yeast exhibits diverse physiological activities, including inhibiting tyrosinase inhibition activity and melanin production to reduce skin aging [1] and antioxidant activity to remove the free radicals and reactive oxygen species formed in the body [4]. Additionally, yeast inhibits the angiotensin-converting enzyme (ACE), vital in maintaining blood pressure [5]. Several yeast strains isolated from naturally fermented foods have been utilized as starters or co-starters for producing functional foods at an industrial site [6].

*Saccharomyces cerevisiae* is a representative Generally Recognized as Safe (GRAS) microorganism that is easy to culture due to its rapid growth rate. Furthermore, it is used in alcoholic beverage production and traditional fermented food manufacturing as it possesses effective alcohol degradation abilities [7]. Several studies established the functional characteristics of yeast in fermented products; yeast can improve folic acid levels [8,9,10,11], increase antioxidant properties [12,13,14], enhance microbiota [10,15,16,17], and increase isoflavone and selenomethionine content [18,19,20]. However, studies have primarily focused on the functional characteristics of *S. cerevisiae*, with limited research being conducted on other types of yeast.

Non-*Saccharomyces* yeast (i.e., non-conventional yeast) is being studied for its applicability to various fields such as fermented food, the brewing industry, and greenery fermentation [21,22,23,24]. In particular, *Wickerhamomyces anomalus* and *Torulaspora delbrueckii* are known as the most promising yeasts in the brewing process of beer and wine, and are applied as innovative tools for manufacturing functional alcoholic beverages, such as providing various aroma profiles and reducing the final ethanol content [22,23,24,25,26].

A culture-independent diversity analysis of traditional Chinese glutinous rice wine revealed the presence of *Pichia*, *Candida*, *Cryptococcus*, *Rhodotorula*, *Sporobolomyces*, and *Rhodosporidium* [27]. Non-*Saccharomyces* yeast, including *Hanseniaspora*, *guilliermondii*, and *Pichia membranifaciens* were the dominant species in cocoa fermentation in Ghana [28]. 

Based on potential probiotic properties, a number of non-*Saccharomyces* yeasts, such as *Pichia kudriavzevii*, *Lachancea thermotolerans*, *Candida vini*, *Zygosaccharomyces bailii*, and *Saccharomyces boulardii*, were evaluated for their applicability as probiotics, including safety and functionality, from a health functional point of view [29,30]. The antagonistic action of the pathogen *Candida albicans*, which can be applied to the pharmaceutical industry, was confirmed [31]. Based on the characteristics and functional aspects of probiotics such as antibacterial activity and the bioprotection and biocontrol of non-*Saccharomyces* yeast, it can be used as a functional material as well as in food production [31,32].

Recently, non-*Saccharomyces* yeasts (non-GRAS) have been utilized in various biotechnological applications [33]. Non-*Saccharomyces* yeasts are currently being used as hosts for the expression of proteins, biocatalysts, and multiple enzymatic pathways to synthesize fine chemicals and low molecular weight compounds of medicinal and nutritional importance. Additionally, they are critical in agriculture as a biological control mechanism, a bioremediation tool, and an environmental quality indicator. Therefore, to elucidate the utility and advantages of yeast, including *Saccharomyces*, we investigated the effects of yeast strains on the production of nitric oxide (NO), NF-κB, and cytokines (inflammatory mediators). Additionally, we provide novel insights into yeast by evaluating skin whitening, antioxidant, antihypertensive, and antibacterial activities.

## 2. Materials and Methods

### 2.1. Isolation and Identification of Strains

To isolate the yeast, 1 g of the sample was suspended in 9 mL of sterile physiological saline and characterized wild-type yeasts, traditional *nuruk*, and *doenjang* from four regions in Korea (the provinces of Gangwon-do, Gyeongsangnam-do, Gyeongsangbuk-do, and Chungcheongbuk-do) were collected (Table 1). Yeast extract peptone dextrose agar (YPD, BD DIFCO, Mississauga, ON, Canada) medium supplemented with 1% penicillin-streptomycin (Gibco-BRL, Waltham, MA, USA) was used to inhibit the growth of mold in the process of isolation, and the supernatant was subjected to a 10-fold serial dilution. Approximately 100 μL of the sample was spread on YPD solid medium and incubated at 25 °C for 48 h to isolate a single colony.

The genomic DNA of the isolates was extracted using an InstaGeneTM Matrix (Bio-Rad, Hercules, CA, USA) according to the manufacturer’s instructions. The amplification of the 18S rRNA gene was performed with universal primers NS1 (5′-GTA GTC ATA TGC TTG TCT C-3′) and NS8 (5′-TCC GCA GTT CAC CTA CGG A-3′), using Axen H Taq PCR Master Mix (Macrogen, Seoul, Republic of Korea) in a DNA Engine Tetrad 2 Peltier Thermal Cycler (Bio-Rad, Hercules, CA, USA) according to the manufacturer’s instructions. The PCR mixtures were preheated for 5 min at 95 °C and amplified using 35 cycles of 0.5 min at 95 °C, 0.5 min at 55 °C, and 0.5 min at 72 °C. The PCR products were purified, analyzed, and sequenced using the two primers described above. Sequencing was performed using the Big Dye Terminator Cycle Sequencing Kit v.3.1 (Applied BioSystems, Waltham, MA, USA), and sequencing products were resolved on an Applied Biosystems 3730XL automated DNA sequencing system (Applied BioSystems). The nucleotide sequence primary analysis was performed using the web-hosted BLAST algorithm with the NCBI database. The MEGA 7 software was used to construct the selected phylogenetic analysis. *Saccharomyces cerevisiae* Fermivin (Saint-Denis, France) was purchased from Samkwang Industrial Co., Ltd. (Seoul, Republic of Korea) and used as a control strain.

### 2.2. Cell Culture and Cell Viability

RAWBlue™ cells (InvivoGen, San Diego, CA, USA) were cultured and maintained at 37 °C in 5% CO_2_ and 95% air in Dulbecco’s modified Eagle’s medium (DMEM, Gibco-BRL, Gaithersburg, MD, USA) containing 10% fetal bovine serum (FBS, Gibco-BRL), 1% penicillin-streptomycin (Gibco-BRL), and 1% Zeocin™ (InvivoGen) [34]. A subculture was performed every two days. Next, the positive control was treated with 100 ng/mL lipopolysaccharide (LPS, *Escherichia coli* O11:B4, Sigma Chemical Co., Saint Louis, MO, USA).

The viability of the macrophages was measured using the 3-(4,5-dimethylthiazole-2-yl)-2,5-diphenyl-tetrazolium bromide (MTT, Sigma-Aldrich Co., St. Louis, MO, USA) reduction method [35]. RAWBlue™ cells were cultured in 96-well plates (Corning^®^ 3369, 96 Well EIA/RIA assay microplate, Corning, New York, NY, USA) at 1 × 10^5^ cells/mL and 37 °C under 5% CO_2_ for 24 h. Yeast strains at 25, 50, and 100 CFU/macrophage concentrations were inoculated, and 10 ng/mL of LPS was used as a positive control. Each plate was cultured at 37 °C and 5% CO_2_ for 24 h to perform the MTT assay. After the cells were treated with yeast strains for 24 h, 200 μL of MTT (5 mg/mL in Dulbecco’s phosphate-buffered saline (D-PBS, Gibco, Grand Island, NY, USA)) was added to each well and incubated for an additional 4 h at 37 °C [24]. After solubilization of the purple-blue MTT formazan crystals, which were formed in viable cells overnight, 200 μL/well of dimethyl sulfoxide (DMSO, Sigma Chemical Co., USA) was added, and the absorbance was measured with a microplate reader (SpectraMax M2, Molecular Devices, Sunnyvale, CA, USA) at 540 nm with DMSO as the blank. The cell viability was expressed as a percentage (%) of the absorbance of each sample relative to the absorbance of the negative control group.
$$Cell\;viability\left(\%\right)=\frac{Absorbance\;({OD}_{540})\;of\;samples}{Absorbance\;({OD}_{540})\;of\;blank}\times 100$$

### 2.3. Activation of NF-κB in LPS-Stimulated RAWBlue™ Cell

The immune response induced by activated macrophages was investigated by measuring NF-κB activation [36]. The macrophages used in this experiment were NF-κB/AP-1 reporter cell lines derived from mouse RAW 264.7 macrophages containing the NF-κB/AP-1-inducible SEAP reporter gene. It is activated upon stimulation of the immune response, and the color change from pink to blue by the indicator (Quanti Blue™, InvivoGen) indicates NF-κB activation [37]. RAWBlue™ cells were cultured in 96-well microplates at a concentration of 2.5 × 10^6^ cells/mL for 24 h, and then treated with yeast strains at concentrations of 25, 50, and 100 CFU/macrophage and further cultured for 24 h. The supernatant was collected and centrifuged at 10,000 rpm for 10 min to remove the remaining cells. After mixing 20 μL of the supernatant with 200 μL of Quanti Blue™ and allowing it to react in the dark for 10 min, the degree of NF-κB activation was measured by reading the absorbance at 650 nm using a microplate reader (SpectraMax M2, Molecular Devices, Sunnyvale, CA, USA).

### 2.4. Nitric Oxide and Cytokine Production of Yeast Strains

The culture supernatant of the RAWBlue™ cells and the amount of nitric oxide (NO) produced were measured to confirm the anti-inflammatory activity of the yeast strains [38]. First, RAWBlue™ cells at a concentration of 2.5 × 10^6^ cells/mL were cultured for 24 h and treated with yeast strains at concentrations of 25, 50, and 100 CFU/macrophage and further cultured for 24 h to obtain a supernatant. Next, 50 μL of the supernatant was mixed with 100 μL of the reagent prepared by mixing equal amounts of Griess reagent (Promega, Madison, WI, USA) I (sulfanilamide solution) and II (NED solution). The reagents were reacted at room temperature for 10 min, and the absorbance (540 nm) was measured using a microplate reader. The NO concentration was calculated using a standard curve for sodium nitrite.

The production of cytokines (IL-1β, IL-6, and TNF-α) was measured from the culture medium obtained after processing RAWBlue™ cells and yeast strains. RAWBlue™ cells were dispensed into 96-well microplates at a concentration of 2.5 × 10^6^ cells/mL and cultured for 24 h; these were treated with yeast strains at concentrations of 25, 50, and 100 CFU/macrophage and further cultured for 24 h, after which the supernatant was collected. Cytokine production was measured by enzyme-linked immunosorbent assay (ELISA) using the mouse IL-1β Uncoated ELISA Kit (Invitrogen, Carlsbad, CA, USA), mouse IL-6 Uncoated ELISA Kit (Invitrogen), and mouse TNF-α Uncoated ELISA Kit (Invitrogen) according to the manufacturer’s instructions. Next, 100 μL of the supernatant was added to each of the 96-well microplates coated with cytokine antibodies and allowed to react at room temperature for 2 h. Next, the supernatant was removed and washed seven times or more with PBS containing 20% Tween 20 (Sigma Chemical Co.) as a washing buffer. After adding the detection antibody solution, the horseradish peroxidase (HRP) enzyme combined with avidin was added and reacted at room temperature for 15 min. As a substrate for the HRP enzyme, 3,3′,5,5′-tetramethylbenzidine solution (TMB, Sigma-Aldrich, St. Louis, MO, USA) was added and reacted to confirm the color change. When cytokines are produced in a sample, a color change occurs; the proportion of cytokines produced was measured by observing the color change. Stop solution (1 M H_2_SO_4_, Sigma-Aldrich) was added to terminate the reaction between the HRP enzyme and the TMB substrate, and the absorbance was measured at 450 and 570 nm using a microplate reader. The data were analyzed by subtracting the 570 nm value from the 450 nm value.

### 2.5. Tyrosinase Inhibition Activity of Yeast Strains

To evaluate extracellular functional characteristics, yeast strains adjusted to an absorbance value of 0.5 at 660 nm were inoculated into 1% YPD broth and cultured at 25 °C for 5 days. The supernatant obtained by the centrifugation of the culture medium at 10,000 rpm for 10 min was used in the experiment [1]. First, the in vitro tyrosinase inhibition activity of yeast strains was assessed using a tyrosinase inhibitor screening assay kit (Abcam, Waltham, MA, USA) to identify potential producers of whitening tyrosinase inhibitors. The assay was performed following the manufacturer’s instructions. Next, 20 μL extracellular yeast supernatant and the control were dispensed into a 96-well microplate, and 50 μL of tyrosinase enzyme solution (48 μL tyrosinase assay buffer and 2 μL tyrosinase enzyme) was added and incubated at 25 °C for 10 min. Next, 30 μL of the tyrosinase substrate mixture was dispensed, and the absorbance was measured at 510 nm and 10 min intervals for 60 min.

$$Relative\;tyrosinase\;inhibition\;activity\left(\%\right)=\frac{A-B}{A}\times 100$$
A: Slope of tyrosinase solution without samples; B: Slope of tyrosinase solution with samples.

### 2.6. Antioxidant Activity of Yeast Strains

The antioxidant activity of the extracellular supernatant of the isolates was analyzed by measuring the DPPH (2,2-diphenyl-1-picrylhydrazyl) and the ABTS (2,2-azino-bis (3-ethylbenzothiazoline-6-sulfonic acid)) free radical scavenging activities. For the DPPH free radical scavenging activity, the OxiTecTM DPPH antioxidant assay kit (BIOMAX, Seoul, Republic of Korea) was used. After dispensing 20 μL of extracellular yeast supernatant in a 96-well plate, 80 μL of assay buffer was dispensed. The DPPH working solution (100 μL) was dispensed into each well; the reaction was performed at room temperature for 30 min by blocking light, and the absorbance was measured at 517 nm [1].

The ABTS antioxidant assay kit (Zenbio, Durham, UK) was used to verify the antioxidant activity through ABTS free radical scavenging activity. First, 80 μL assay buffer was added to 1.5 mM Trolox standard, mixed to prepare 300 μM Trolox standard solution, and then serially diluted to 4.688 μM in a 96-well microplate. Next, 80 μL of the extracellular yeast supernatant was dispensed into each well, 100 μL of assay buffer was added, and 20 μL of myoglobin working solution was dispensed. After adding 100 μL ABTS solution, the mixture was left at room temperature for 5 min using a plate shaker. Finally, after dispensing 50 μL of the stop solution into each well to terminate the reaction, the absorbance was measured at 405 nm.

$$DPPH\;and\;ABTS\;radical\;scavenging\;activity\left(\%\right)=\left(1-\frac{A}{B}\right)\times 100$$
A: Absorbance of DPPH or ABTS solution with samples at 517 or 405 nm; B: Absorbance of DPPH or ABTS solution without samples at 517 or 405 nm.

### 2.7. Antihypertensive Activity of Yeast Strains

The antihypertensive activity was evaluated by screening for angiotensin-converting enzyme (ACE) inhibition and fibrinolytic activity. ACE inhibition screening of isolates was performed using an ACE inhibition screening kit (Kamiya Biomedical Company, Washington, DC, USA). First, 20 μL of extracellular supernatant of the isolates was added; then, equal amounts of substrate buffer and enzyme working solution were mixed and left to react at 37 °C for 1 h. Next, 200 μL of indicator working solution was added and incubated for 10 min at room temperature, and the absorbance (450 nm) was measured. As a positive control, 1% captopril was used, and the ACE inhibition activity was calculated by comparison with the captopril standard.

$$ACE\;inhibition\left(\%\right)=\frac{A-B}{A-C}\times 100$$
A: Absorbance of blank 1 (no ACE inhibition; positive control); B: Absorbance of the sample; C: Absorbance of blank 2 (reagent blank).

Fibrinolytic activity was measured by tyrosine quantification using a partially modified method [39]. After the 0.5% fibrinogen solution was prepared, 250 μL of extracellular yeast culture supernatant was added to 1.5 mL and reacted at 40 °C for 10 min. The reaction was stopped by adding 1.5 mL of 0.4 M trichloroacetic acid solution (T9159, Sigma-Aldrich), and the coagulum was filtered through a paper filter. After adding 1 mL of 6% sodium carbonate (Na_2_CO_3_, Sigma-Aldrich) to 400 μL of the filtrate, 200 μL of 1 N Folin-Ciocalteu reagent (Sigma-Aldrich) was added to develop the color. The amount of tyrosine released from the fibrinogen was quantified by measuring the absorbance at 660 nm. A standard graph for 1 U/mL plasmin (P1867, Sigma-Aldrich) with three replications of different plasmin concentrations against tyrosine released owing to fibrinolytic activity was prepared. The fibrinolytic activity was expressed relative to that of 1 U/mL plasmin.

### 2.8. Antibacterial Activity of Yeast Strains

The antibacterial activity of yeast strains against pathogenic bacteria was tested by partially modifying this method [40,41]. Pathogens (*Bacillus cereus* KACC 10004, *Staphylococcus aureus* ATCC 6538, *Escherichia coli* KCTC 1309, and *Salmonella typhimurium* KCTC 41028) and yeast strains adjusted to an absorbance value of 0.5 at 660 nm were inoculated at 1% (*v*/*v*) in 2 mL of tryptic soy broth (TSB, Difco Laboratories Inc., Detroit, MI, USA) and cultured with shaking at 37 °C for 24 h. The culture medium was diluted to 10^3^–10^7^, and 30 μL was spread on tryptic soy agar (TSA, Difco), incubated at 37 °C for 24 h, and bacterial colonies were counted.

### 2.9. Statistical Analysis

All experiments were repeated three times, and the results are expressed as mean ± standard deviation. One-way analysis of variance, followed by Duncan’s multiple range test, was used to evaluate the significance of the differences between the averages. Statistical significance was set at *p* < 0.05. All statistical analyses were performed using SPSS version 17.0 (SPSS Inc., Chicago, IL, USA).

## 3. Results and Discussion

### 3.1. Isolation and Characterization of Yeast Strains from Traditional Fermented Food in Korea

Ten yeast strains comprising three (*Debaryomyces hansenii*, *Starmerella geochares* (*Candida sorbosivorans*), and *Millerozyma farinose* (*Pichia sorbitophila*) isolated from doenjang with high salinity), and seven strains (one *Millerozyma farinose* (*Pichia sorbitophila*), one *Hyphopichia burtonii*, two *Saccharomyces cerevisiae*, one *Wickerhamomyces anomalus*, one *Saccharomycopsis fibuligera*, and one *Nakaseomyces glabratus* isolated from wheat *nuruk* (Table 1)) were isolated from four regions in Korea (Gangwon-do, Gyeongsangnam-do, Gyeongsangbuk-do, and Chungcheongbuk-do). The taxonomic status of ten isolates was confirmed in the 18S rRNA-based phylogenetic tree; nine isolates were phylogenetically closely located, except for *Starmerella geochares* D6–P9 (Figure 1). Currently, the Korea Ministry of Food and Drug Safety (https://www.foodsafetykorea.go.kr/, accessed on 13 February 2023) permits the limited use of *Debaryomyces hansenii* for dairy product manufacturing. *Saccharomyces cerevisiae* is acceptable for use in food; only *Candida utilis* (*Pichia jadinii*) is permissible from the *Candida* and *Pichia* genera.

### 3.2. Effect of Yeast on Cell Viability and NF-κB Activation

In the LPS-stimulated RAWBlue™ macrophages, the cell viability, according to the concentration of the yeast strain, was investigated using the MTT assay (Figure 2a). When the cell viability of the control group (not treated with any drug) was 100%, the cell viability of the LPS-stimulated group decreased to 85%. However, when the yeast strains were treated at 25, 50, and 100 CFU/macrophage concentrations, the cell viability increased as the yeast strain concentration decreased. In particular, the cell viability was improved, similar to that of the control group, at concentrations of 25–50 CFU/macrophage.

When macrophages are stimulated with LPS, various inflammatory factors increase; NF-κB is an inflammatory response-related transcription factor produced by lymphocyte and macrophage activation. To evaluate the anti-inflammatory activity of yeast in LPS-stimulated RAWBlue™ cells, the NF-κB signaling pathway was investigated. The NF-κB levels were barely detectable in unstimulated RAWBlue™ cells (Figure 2b). On the other hand, the exposure of cells to LPS dramatically increased the NF-κB. The D5–P5, D6–P9, and D10–P12 strains inhibited NF-κB activity at concentrations of 25, 50, and 100 CFU/macrophage. The NR4 and NR5 strains exhibited NF-κB inhibition activity at 25–50 CFU/macrophage.

### 3.3. Effect of Yeast on Nitric Oxide Production and Expression of iNOS and COX-2 in LPS-Stimulated RAWBlue™ Cells

To evaluate the anti-inflammatory effect of yeast, we examined the effect of yeast on NO synthesis in LPS-stimulated RAWBlue™ cells. The addition of yeast (25, 50, and 100 CFU/macrophage) to LPS-stimulated RAWBlue™ cells decreased the amount of NO in the cell culture supernatant in a concentration-dependent manner (Figure 3a). In the case of the D6–P9 strain, the production of NO increased at a concentration of 25 CFU/macrophage and decreased as the concentration of the yeast strain increased. We further confirmed whether yeast-induced NO inhibition was induced by the reduced expression of iNOS and COX-2 mRNA in yeast-stimulated cells compared to LPS-stimulated cells (Figure 3b,c). A decrease in iNOS expression in many isolates was observed after 100 CFU/macrophage treatment. However, treatment with D5–P5 and D6–P9 did not affect iNOS expression. In contrast to iNOS, COX-2 expression decreased when the control Fermivin strain was used; this indicates that the inhibition of NO production by yeast in LPS-stimulated RAWBlue™ cells results from the inhibition of iNOS or COX-2 mRNA expression, depending on the strain [42].

### 3.4. Effect of Yeast on Pro-Inflammatory Cytokine Expression in LPS-Stimulated RAWBlue™ Cells

Pro-inflammatory cytokines (IL-1β, IL-6, and TNF-α) are essential in increasing and aggravating the inflammatory response [43]; the decreased expression of cytokines is crucial to avoid further inflammatory responses [42]. Therefore, the amount of cytokines secreted from LPS-stimulated RAWBlue™ cells was analyzed by ELISA. RAWBlue™ cells were stimulated with LPS and simultaneously treated with yeast strains at 25, 50, and 100 CFU/macrophage. IL-1β, IL-6, and TNF-α were barely detectable in unstimulated RAWBlue™ cells (Figure 4a–c); in contrast, the exposure of cells to LPS dramatically increased the levels of IL-1β, IL-6, and TNF-α. Treatment with D5–P5, D6–P9, and D10–P12 suppressed the IL-1β, similar to that in LPS-stimulated cells. D5–P5, D6–P9, D10–P12, and NR5 strains suppressed IL-6 levels at some concentrations, and NR4 strains suppressed IL-6 levels at 25, 50, and 100 CFU/macrophage. TNF-α was suppressed in all isolates except D5–P5.

Although there were differences among the strains since the levels of pro-inflammatory cytokines decreased after yeast treatment, RT-PCR was used to measure mRNA expression. The three pro-inflammatory cytokines were rarely expressed in unstimulated RAWBlue™ cells (Figure 4d–f). In contrast, the exposure of cells to LPS induced the expression of pro-inflammatory-cytokine-related mRNA. Compared to LPS-stimulated cells, all isolates, except Fermivin, failed to inhibit IL-1β transcription. The transcription of IL-6 and TNF-α was suppressed when the yeast strain was treated; however, there was a difference due to the strain concentrations, indicating that the yeast strain suppressed the pro-inflammatory cytokines IL-6 and TNF-α at the mRNA level.

### 3.5. Evaluation of Extracellular Tyrosinase Inhibition Activity

Melanin, distributed in hair and skin, is a protein complex produced by melanocytes located in the basal layer of the skin [44]. It is used as an indicator to verify the whitening efficacy by measuring the tyrosinase enzyme involved in melanin production [45]. As a result of confirming the whitening effect by measuring the extracellular tyrosinase inhibition activity of the isolates, five isolates demonstrated <10% activity (Table 2). Reportedly, the extracellular activity of wild-type yeast was 10–20%, indicating a similar trend to the five isolates in this study [46]. In contrast, the intracellular activity was approximately 20% higher than the extracellular activity [1,46]. Therefore, wild yeast possesses tyrosinase inhibitory activity, but does not secrete it extracellularly; further research is needed.

### 3.6. Evaluation of Extracellular Antioxidant Activity

Aging, arteriosclerosis, and skin diseases occur due to damage to or the destruction of cells [1] caused by reactive oxygen species (such as superoxide anions, hydrogen peroxide, and hydroxyl radicals). Specific yeasts are known to be inhibited by reactive oxygen species [47]. In evaluating DPPH and ABTS radical scavenging activities, NR4 exhibited the highest radical scavenging activity among the isolates (Table 2). The five isolates exhibited approximately 40% and 90% or more DPPH and ABTS radical scavenging activity, respectively. Although there were slight differences according to the two methods, the tendency between the strains was similar. Yeast isolated from virgin olive oil exhibited a DPPH radical scavenging activity of 40–60% [48]. The intracellular ABTS radical scavenging ability of *Saccharomyces cerevisiae* strains demonstrated 30–80% activity depending on the incubation time [49]. The DPPH and ABTS free radical scavenging activities of yeast were demonstrated in an in vitro test with five strains of yeast isolated from traditional Korean fermented foods. The antioxidant activity of yeast is mainly due to the content of (1,3), (1,6)-β-D-glucan, and protein fractions located in the cell wall [49,50,51]. Glucans from different sources and molecular weights have different antioxidant activities [52]. The various specific scavenging activities of the yeast strains listed in Table 2 support these results.

### 3.7. Evaluation of Extracellular Antihypertensive Activity

Fermented foods containing ACE-inhibitory peptides offer antihypertensive benefits [6,53]. In the evaluation of extracellular ACE inhibition activity, five isolates demonstrated an average activity of 73.49% compared to the positive control (Table 2). The ACE inhibition activity of D5–P5 was the highest at 91.16%, and the activity of D10-P12 was the lowest at 54.01%. Yeast isolated from fermented milk products produced ACE inhibition peptides upon the hydrolysis of milk protein [54,55]. In addition, milk fermented with a single culture of the yeast strain produced higher ACE inhibition peptides than the mixed culture fermented with lactic acid bacteria [54].

Cerebrovascular disease is caused by blood clots obstructing the blood flow [56]. The formation of blood clots results from the conversion of fibrinogen (a plasma protein) to fibrin by activated thrombin [57]. When excessive blood clots are in the blood vessels or the physiological functions related to thrombolysis are abnormal, thrombosis may occur; consequently, oxygen and nutrients cannot be supplied to specific tissues, causing severe disorders [58]. All strains demonstrated higher fibrinolytic activity than the positive control, with an average activity of 126.20% compared to the positive control (100% activity at 1 U/mL plasmin). The fibrinolytic activity of D6–P9 was the highest (133.21%), and the activity of NR5 was the lowest (118.39%). The extracellular antihypertensive activity slightly differed between the two methods; however, all strains exhibited relatively high antihypertensive activity, suggesting it can be used as a functional fermented food (using yeast substances with antihypertensive activity), a healthy functional food, and antihypertensive medicine.

### 3.8. Antibacterial Activity of Yeast Strains

Figure 5 illustrates the antibacterial activity of the five isolates against two pathogenic Gram-positive bacteria (*Bacillus cereus* and *Staphylococcus aureus*) and two pathogenic Gram-negative bacteria (*Escherichia coli* and *Salmonella typhimurium*). Compared to the case where only pathogenic bacteria were cultured with yeast strains, five isolates exhibited significant antibacterial activity, resulting in growth inhibition against Gram-negative pathogenic bacteria. Isolates reduced the growth of *S. typhimurium* by 0.5–0.7 log, and the growth of *E. coli* by 1.3–1.6 log. However, none of the five isolates demonstrated growth inhibition against pathogenic Gram-positive bacteria. Purified *Pichia kudriavzevii* RY55 killer toxin exhibits potent antibacterial activity against pathogenic bacteria such as *Escherichia coli*, *Enterococcus faecalis*, *Klebsiella* sp., *Staphylococcus aureus*, *Pseudomonas aeruginosa*, and *Pseudomonas alcaligenes* [59]. In addition, the killer toxins of *Hansenula*, *Kluyveromyces*, *Candida*, and *Saccharomyces* possess potential growth-inhibitory activity against Gram-positive bacteria [60,61]. These findings support the results of *Debaryomyces hansenii* D5–P5, *Starmerella geochares* D6–P9, *Millerozyma farinosa* D10–P12, *Hyphopichia burtonii* NR4, and *Saccharomyces cerevisiae* NR5 strains, which specifically inhibited the growth of Gram-negative pathogenic bacteria. Yeasts with antibacterial activity may inhibit the growth of bacteria that cause food spoilage and pathogenicity during fermentation. Therefore, yeast application in the food industry has been suggested [62].

## 4. Conclusions

To evaluate the functional characteristics of new wild-type yeasts, ten strains were isolated from traditional fermented foods in Korea; their taxa were confirmed using an 18S rRNA-based phylogenetic tree. Among them, those exhibiting high alcohol production exhibited similar improvements in RAWBlue™ cell viability at 25–50 CFU/macrophage concentrations compared to the control RAWBlue™ cell group. Five isolates inhibited NF-κB activity in LPS-stimulated RAWBlue™ cells, particularly the NR4 strain, by 72% compared to LPS. The production of NO in yeast and LPS-stimulated RAWBlue™ cells decreased in a concentration-dependent manner due to the inhibition of iNOS or COX-2 mRNA expression in a strain-specific pattern. The production of the anti-inflammatory cytokines IL-1β, IL-6, and TNF-α suppressed pro-inflammatory cytokines at the mRNA level in yeast and LPS-stimulated RAWBlue™ cells, especially for the NR4 strain.

The five isolates demonstrated low extracellular tyrosinase inhibition activity, while NR4, NR5, and D5–P5 exhibited high antioxidant activity. The radical scavenging activity of yeast derived from various isolates has been reported [48,49,52], demonstrating the high antioxidant activity of the yeast. In addition, NR4, D5–P5, and D6–P9 exhibited high extracellular antihypertensive and antibacterial activity against pathogenic Gram-negative bacteria. As awareness of the role of yeasts in functional food production increases, the selection of potential yeast strains to improve the functional properties of a product has become an important criterion [6]; this suggests that yeasts with antioxidant, antihypertensive, and antibacterial activities could improve the functional properties of fermented foods in the food industry. In addition, yeast is highly productive, and the cell wall components can be a potential source of bioactive molecules that provide functional properties to products [6]. Yeast has high potential as a functional material; however, additional genetic and dietary safety verification studies are needed for non-*Saccharomyces cerevisiae* strains not listed as raw food materials. The health benefits and potential applications of functional materials obtained through natural methods have generated increasing interest [63]. Microbial-derived functional materials are considered a valuable alternative because plant extraction is sometimes expensive, and the quality of the obtained product is determined by the batch of raw materials. Exopolysaccharides, lipids, and carotenoids produced by yeast exhibit physicochemical and rheological properties useful in food production and cosmetic and pharmaceutical industries [63,64,65]. Based on the various physiological functions of yeast strains, they are applicable in food, feed, and medicine.

## Figures and Tables

**Figure 1 microorganisms-11-01503-f001:**
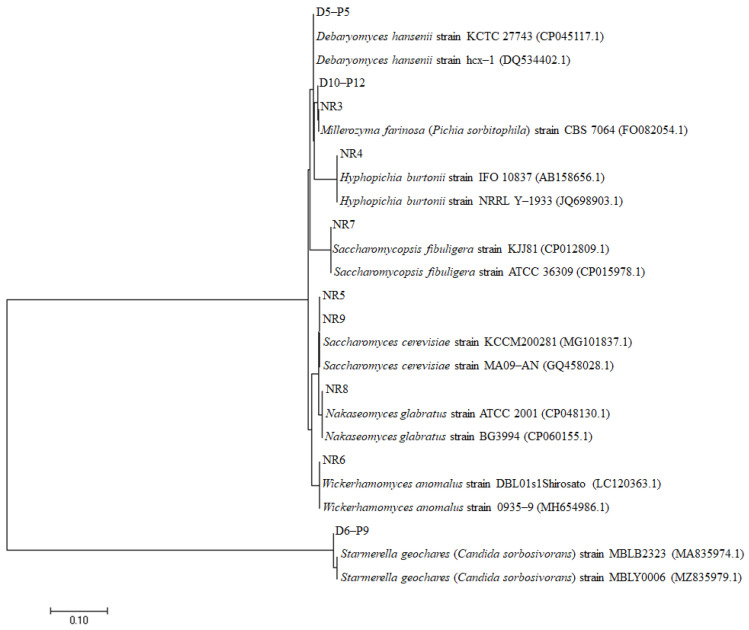
Phylogenetic tree of 10 yeast strains isolated from traditional *doenjang* and *nuruk* in various regions in Korea. A maximum-likelihood phylogenetic tree was constructed using fifteen long 18S rRNA gene sequences from *Debaryomyces hansenii*, *Millerozyma farinose*, *Hyphopichia burtonii*, *Saccharomycopsis fibuligera*, *Saccharomyces cerevisiae*, *Nakaseomyces glabratus*, and *Starmerella geochares* sequences from GenBank. The accession numbers of each strain in GenBank are presented in parentheses.The isolate was inoculated into 1% YPD broth containing 25% glucose (*v*/*v*) and cultured at 25 °C for 7 days to evaluate the sugar consumption and alcohol production of the yeast strains. Among the ten isolates, D10–P12 exhibited the highest alcohol production, followed by NR4, D6–P9, D5–P5, and NR5 (Table 1). Therefore, the functional characteristics of the five selected yeast strains were evaluated.

**Figure 2 microorganisms-11-01503-f002:**
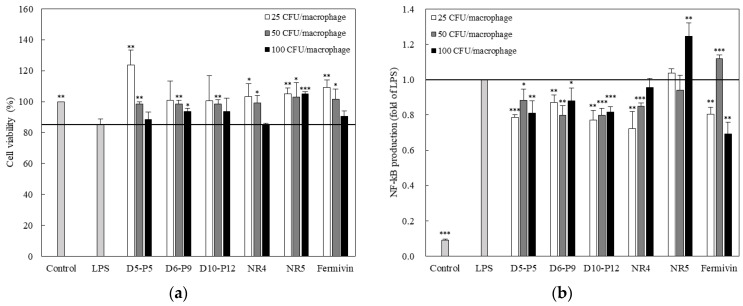
Dose-dependent cell viability in yeast strains (**a**). RAWBlue™ (1 × 10^5^/well plate) with or without LPS (10 ng/mL) in the presence or absence of yeast at indicated doses. Cell viability was measured by MTT assay. Effects of yeast strains on the production of NF-κB (**b**) in LPS (100 ng/mL)-induced RAWBlue™ cell. The data represent the mean ± SD of triplicate experiments. ** p* < 0.05, *** p* < 0.01, and **** p* < 0.001 compared with LPS-only-treated cells.

**Figure 3 microorganisms-11-01503-f003:**
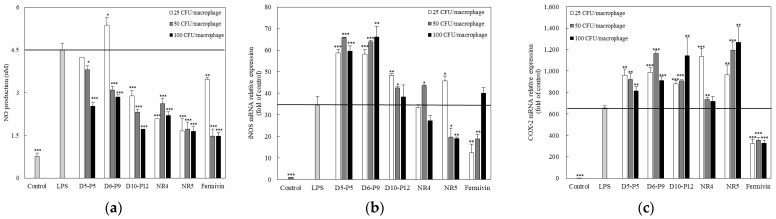
Effects of yeast strains on LPS (100 ng/mL)-stimulated NO production in RAWBlue™ cells. Nitrite level (**a**) in the culture medium was measured using Griess Regent. iNOS (**b**) and COX-2 (**c**) mRNA expression was detected by RT-PCR. The data represent the mean ± SD of triplicate experiments. ** p* < 0.05, *** p* < 0.01, and **** p* < 0.001 compared with LPS-only-treated cells.

**Figure 4 microorganisms-11-01503-f004:**
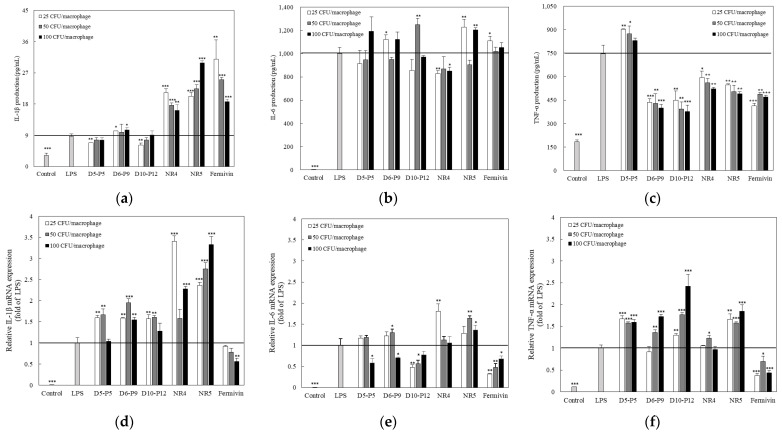
Effects of yeast strains on LPS (100 ng/mL)-induced pro-inflammatory cytokines in RAWBlue™ cells. IL-1β (**a**), IL-6 (**b**), and TNF-α (**c**) levels in the culture medium were measured using ELISA. mRNA levels of IL-1β (**d**), IL-6 (**e**), and TNF-α (**f**) were determined using RT-PCR. The data represent the mean ± SD of triplicate experiments. ** p* < 0.05, *** p* < 0.01, and **** p* < 0.001 compared with LPS-only-treated cells.

**Figure 5 microorganisms-11-01503-f005:**
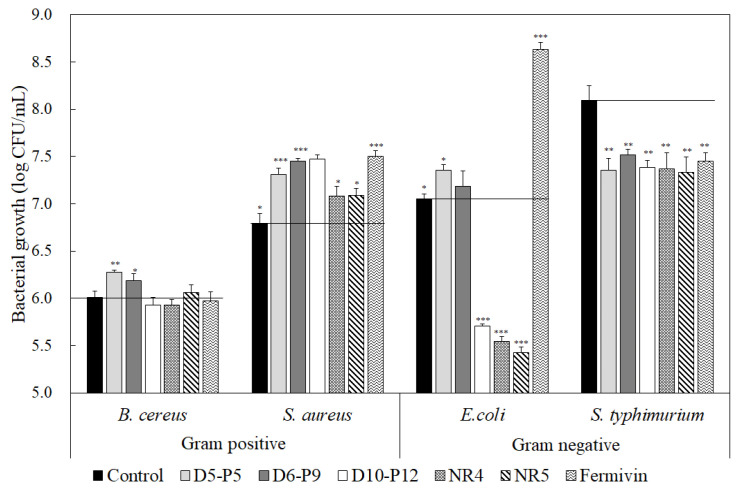
Antibacterial activity of the yeast isolates against *Bacillus cereus (B. cereus)* KACC 10004, *Staphylococcus aureus (S. aureus)* ATCC 6538, *Escherichia coli (E. coli)* KCTC 1309, and *Salmonella typhimurium (S. typhimurium)* KCTC 41028. Control represents the result of inoculating bacteria only. The data represent the mean ± SD of triplicate experiments. ** p* < 0.05, *** p* < 0.01, and **** p* < 0.001 compared with the bacteria-only-treated control.

**Table 1 microorganisms-11-01503-t001:** Yeast strains isolated from traditional *doenjang* and *nuruk* in various regions of Korea.

Origin	Region (Province, Korea)	Species	Strain	Alcohol Production (%)
*Doenjang* (15% NaCl)	Gangwon-do	*Debaryomyces hansenii*	D5–P5	11.33
		*Starmerella geochares* (*Candida sorbosivorans*)	D6–P9	11.48
	Gyeongsangnam-do	*Millerozyma farinose* (*Pichia sorbitophila*)	D10–P12	12.10
Wheat *nuruk*	Gyeongsangnam-do	*Millerozyma farinose* (*Pichia sorbitophila*)	NR3	11.41
	Chungcheongbuk-do	*Hyphopichia burtonii*	NR4	11.54
		*Saccharomyces cerevisiae*	NR5	11.25
		*Wickerhamomyces anomalus*	NR6	0.68
		*Saccharomycopsis fibuligera*	NR7	7.55
	Gangwon-do	*Nakaseomyces glabratus*	NR8	3.27
	Gyeongsangbuk-do	*Saccharomyces cerevisiae*	NR9	1.26
Saint-Denis, France	–	*Saccharomyces cerevisiae*	Fermivin ^1^	14.0

^1^ As a control strain, *Saccharomyces cerevisiae* Fermivin (Saint-Denis, France), purchased from Samkwang Industrial Co., Ltd. (Seoul, Republic of Korea), was used.

**Table 2 microorganisms-11-01503-t002:** Comparison of relative biochemical characterization for isolates.

Strain	Tyrosinase Inhibition Activity	Antioxidant Activity		Antihypertensive Activity ^1^	
		DPPH Radical Scavenging	ABTS Radical Scavenging	ACE Inhibition	Fibrinolysis
D5–P5	4.57 ± 1.52 ^2bc^	44.91 ± 2.78 ^a^	98.33 ± 1.1 ^bc^	91.16 ± 0.67 ^b^	124.21 ± 2.98 ^bc^
D6–P9	9.77 ± 2.24 ^a^	42.34 ± 5.09 ^a^	95.87 ± 1.89 ^cd^	77.33 ± 2.24 ^c^	133.21 ± 3.60 ^a^
D10–P12	ND ^3^	43.68 ± 7.17 ^a^	95.00 ± 0.44 ^d^	54.01 ± 8.40 ^e^	125.93 ± 1.19 ^abc^
NR4	3.03 ± 0.05 ^c^	48.00 ± 0.18 ^a^	102.57 ± 0.61 ^a^	72.74 ± 2.50 ^cd^	129.24 ± 4.43 ^ab^
NR5	4.99 ± 0.20 ^bc^	47.32 ± 1.16 ^a^	97.93 ± 2.17 ^bc^	68.54 ± 3.29 ^d^	118.39 ± 6.77 ^c^
Fermivin	6.13 ± 1.26 ^b^	31.65 ± 2.17 ^b^	101.53 ± 1.07 ^a^	77.17 ± 2.16 ^c^	122.36 ± 1.19 ^c^
PC ^4^	10.82 ± 2.93 ^a^	48.77 ± 4.54 ^a^	102.58 ± 1.77 ^a^	100.00 ± 0.40 ^a^	100.00 ± 5.70 ^d^

^1^ Antihypertensive activity was calculated by comparing with each positive control standard. ^2^ The results are expressed as mean ± SD (*n* = 3). As determined by Duncan’s multiple range test, the means of different letters (a–e) with different values are significantly different at *p* < 0.05. ^3^ Not detected. ^4^ Positive control of each treatment is below 0.75 mM Kojic acid for tyrosinase inhibition activity, 0.3 mM Trolox for DPPH radical scavenging activity, 0.05 mM Trolox for ABTS radical scavenging activity, 1% captopril for ACE inhibition activity, and 1 U/mL plasmin for fibrinolytic activity. Abbreviations: DPPH, 2,2-diphenyl-1-picrylhydrazyl; ABTS, 2,2-azino-(3-ethylbenzothiazoline-6-sulfonic acid); ACE, angiotensin-converting enzyme.

## Data Availability

All data are contained within the article.

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
