# Peer review of "Exploring the Health Benefits of Yeast Isolated from Traditional Fermented Foods in Korea: Anti-Inflammatory and Functional Properties of Saccharomyces and Non-Saccharomyces Strains"

_microorganisms, 2023, doi:10.3390/microorganisms11061503_

Round 1

Reviewer 1 Report

The current study explored the anti-inflammatory and functional properties of Saccharomyces and Non-Saccharomyces strains, focusing on the health benefits of yeast isolated from traditional fermented foods in Korea. The whole work is valuable and interesting. There are some minor comments for improving the manuscript:

1. The supplementary material has only one table, which could be included into the main manuscript in order to provide detailed information for the ID of isolated strains. 

2. The 18S sequences should be uploaded onto a common database, in order to allow readers to double check the strain species. 

3. Based on the evolution tree in the Fig.1, the 18S sequences of the isolated strains are very similar. Could authors provide information about how many base pairs are different among these strains, and how to confirm that the species identification is correct?

4. The Fig.3, 4 and 5 are not clear for the X and Y axis. 

Author Response

First of all, thank you for reviewing my paper.

1. The supplementary material has only one table, which could be included into the main manuscript in order to provide detailed information for the ID of isolated strains.

-> Supplementary data (Table 1.) were added to the main manuscript

2. The 18S sequences should be uploaded onto a common database, in order to allow readers to double check the strain species. 

-> Based on the functionality of the investigated yeast, it is currently under review with a plan to deposit a patent strain.

-> After depositing the patent, we will upload it to the common database.

3. Based on the evolution tree in the Fig.1, the 18S sequences of the isolated strains are very similar. Could authors provide information about how many base pairs are different among these strains, and how to confirm that the species identification is correct?

-> The nucleotide sequence obtained by 18S sequencing was allingmented with a program such as clustal Omega to reconfirm some of the nucleotide sequences that did not match, and as a result, it was confirmed that it was not the same strain showing differences in nucleotide sequence between species. In addition, if the entire nucleotide sequence between species is very similar, we want to distinguish them through WGS.

4. The Fig.3, 4 and 5 are not clear for the X and Y axis. 

-> As shown in Fig 2., in the case of concentration, it indicates the amount of cytokine production, so the graph axis was modified to be visually confirmed on the graph. For the graph using the mRNA expression level as the Y-axis, the graph axis was corrected based on the LPS treatment control (Fold of LPS) as 1.0. In Fig5, the y-axis was clearly modified to confirm the degree of inhibition of growth for each bacteria compared to the control.

Reviewer 2 Report

The paper "Exploring the Health Benefits of Yeast Isolated from Traditional Fermented Foods in Korea: Anti-inflammatory and Functional Properties of Saccharomyces and Non-Saccharomyces Strains" propose a study on the benefits of yeast isolated from traditional food for their anti - inflammatory and functional property.The description of methods was clear and exhaustive as results. Overall, the work offers interesting ideas for the study of the beneficial effects that can come from the microbial populations of traditional fermented products, enhancing their functional properties.Overall, the work offers interesting ideas for the study of the beneficial effects that can come from the microbial populations of traditional fermented products, enhancing their functional and antibacterial properties. I would suggest, to the authors, to widen the description and the activities and the benefits of Non - Saccharomyces yeast, topic not yet much discussed, just to emphasize the innovation of the work presented.

Author Response

First of all, thank you for reviewing my paper.

-> I added some features and applications for non-Saccharomyces yeast to the Introduction.

- Non-Saccharomyces yeast used in food and brewing industries - Probiotic yeast related to health functional aspects
